# Antibiotic Allergy De-Labeling: A Pathway against Antibiotic Resistance

**DOI:** 10.3390/antibiotics11081055

**Published:** 2022-08-03

**Authors:** Inmaculada Doña, Marina Labella, Gádor Bogas, Rocío Sáenz de Santa María, María Salas, Adriana Ariza, María José Torres

**Affiliations:** 1Allergy Research Group, Instituto de Investigación Biomédica de Málaga-IBIMA, 29009 Málaga, Spain; inmadd@hotmail.com (I.D.); labellaalvarezmarina@gmail.com (M.L.); gabhdor@hotmail.com (G.B.); rociossm.93@gmail.com (R.S.d.S.M.); mariasalascassinello@hotmail.com (M.S.); a.arizaveguillas@gmail.com (A.A.); 2Allergy Unit, Hospital Regional Universitario de Málaga, 29009 Málaga, Spain; 3Nanostructures for Diagnosing and Treatment of Allergic Diseases Laboratory, Andalusian Center for Nanomedicine and Biotechnology-BIONAND, 29009 Málaga, Spain; 4Departamento de Medicina, Universidad de Málaga, 29009 Málaga, Spain; 5Research Unit for Allergic Diseases, IBIMA-Regional University Hospital of Malaga, Pl. Hospital Civil, 29009 Málaga, Spain

**Keywords:** allergy, drug provocation test, IgE, in vitro test, skin test, T-cell

## Abstract

Antibiotics are one of the most frequently prescribed drugs. Unfortunately, they also are the most common cause for self-reported drug allergy, limiting the use of effective therapies. However, evidence shows that more than 90% of patients labeled as allergic to antibiotics are not allergic. Importantly, the label of antibiotic allergy, whether real or not, constitutes a major public health problem as it directly impacts antimicrobial stewardship: it has been associated with broad-spectrum antibiotic use, often resulting in the emergence of bacterial resistance. Therefore, an accurate diagnosis is crucial for de-labeling patients who claim to be allergic but are not really allergic. This review presents allergy methods for achieving successful antibiotic allergy de-labeling. Patient clinical history is often inaccurately reported, thus not being able to de-label most patients. In vitro testing offers a complementary approach but it shows limitations. Immunoassay for quantifying specific IgE is the most used one, although it gives low sensitivity and is limited to few betalactams. Basophil activation test is not validated and not available in all centers. Therefore, true de-labeling still relies on in vivo tests including drug provocation and/or skin tests, which are not risk-exempt and require specialized healthcare professionals for results interpretation and patient management. Moreover, differences on the pattern of antibiotic consumption cause differences in the diagnostic approach among different countries. A multidisciplinary approach is recommended to reduce the risks associated with the reported penicillin allergy label.

## 1. Introduction

The use of broad-spectrum antibiotics in patients labeled as “antibiotic allergic” is an important contributor to inappropriate antibiotic use, with penicillin allergy being the most common label. Globally, up to 10% of the population is considered to be allergic to penicillins [1]. However, in the vast majority of patients labeled as allergic to antibiotics, mainly penicillins, an allergological study confirming the allergy has not been performed, and, additionally, when the allergological evaluation is carried out, allergy is ruled out in more than 70–80% of adults and in more than 90% of children [2,3]. This is due to the fact that most cases are labeled as antibiotic allergy due to occurrence of delayed benign rashes as a result of viral infections or antibiotic intolerances [1,4]. Specifically in the pediatric population, up to 5% of β-lactam antibiotic prescriptions are associated with rashes [5].

Overdiagnosis of antibiotic allergy, whether real or not, ultimately translates into these patients being more likely to be treated with alternative broad-spectrum antibiotics [6,7,8,9], which are associated with negative clinical consequences such as increased surgical site infections [10], increased antibiotic-resistance and healthcare-associated infections (14.1% more methicillin-resistant *Staphylococcus aureus* and 30.1% more vancomycin-resistant *Enterococcus*), and higher rate of serious infections (23.4% more *Clostridium difficile* infections), which translates into the need for longer hospitalizations than in non-allergic patients [7,9,11,12,13,14,15,16,17,18].

This ultimately represents a higher risk to the patient’s health and higher healthcare expenditures [19,20].

Given the role of broad-spectrum antibiotics in antibiotic resistance, testing for antibiotic allergy is an important pillar of antibiotic stewardship programs. De-labeling antibiotic allergies in the high proportion of patients with suspected antibiotic allergy who are not really allergic is an important first step for decreasing resistance and optimizing medical care [4].

## 2. Classification of Allergic Reactions to Antibiotics

More than 80% of all adverse reactions to antibiotics comprise type A reactions, which are related to the drug’s predictable pharmacological properties. Type B reactions, which are less predictable in relation to dose and pharmacological action and represent true hypersensitivity due to idiosyncratic and individual predisposition, are less common [21,22]. Only when a definite immunological mechanism is demonstrated, these reactions should be termed as allergic [22]. The classification of type B or hypersensitivity reactions to drugs relies on the clinical presentation of typical symptoms and their timing as immediate reactions (IRs), if they occur within 1–6 h after the drug administration, or non-immediate (NIRs), whether they occur with a longer interval, usually after several hours or even days [23,24].

Clinically differentiating an adverse reaction from hypersensitivity to drugs is difficult due to the wide variety of symptoms, with the skin being the organ most frequently involved. In IRs, the severity ranges from urticaria/angioedema to life-threatening anaphylactic shock. In NIRs, urticaria and benign maculopapular rash are the most common manifestations, but severe cutaneous adverse reactions can occur, although they are rare [23]. These comprise acute generalized exanthematous pustulosis (AGEP), toxic epidermal necrolysis (TEN), Stevens–Johnson syndrome (SJS) and drug reaction with eosinophilia and systemic symptoms (DRESS) [25].

The pathogenesis of hypersensitivity reactions to drugs can implicate various immunological mechanisms. Classically, the following classification was described by Gell and Coombs [26] into four types of hypersensitivity reactions described in Figure 1. NIRs are mainly T-cell specific or delayed by the IgG-mediated mechanism [23]. IRs are mainly mediated by IgE antibodies. However, other underlying mechanisms have been recently described, so that the activation of basophil and mast cell may be derived by direct or indirect mechanism, an IgE/IgG or complement-mediated [27]. These recent insights make it more difficult to clinically diagnose an allergic reaction and claim to review the original classification (Figure 1).

## 3. Antibiotics Involved in Allergic Reactions

The most frequent antibiotics involved in hypersensitivity reactions are β-lactam s, representing up to 18% of patients with confirmed reactions to drugs, followed by quinolones (7%), macrolides (2%), metronidazole (1.8%) and other antibiotics, such as clindamycin and sulfonamides, representing less than 1% [2]. Prevalence has changed over the years due to changes in prescribing and consumption patterns, resulting in modifications in patterns of sensitization, as in the case of β-lactam allergic determinants. Therefore, amoxicillin has been gradually replacing benzylpenicillin as the main culprit of allergic reactions and reactions to clavulanic acid have progressively increased in the last few years [28], although amoxicillin is still the drug that most frequently induces reactions [29].

## 4. The Complexity of the Evaluation of Antibiotic Allergy

Predictive models based on the clinical history of patients with suspicions of allergic reactions to β-lactam, as well as clinical decision-making algorithms, have proved to be unable to accurately differentiate between allergic and nonallergic individuals [30,31,32]. Both European and U.S. experts have published statements on general procedures for evaluating HDRs [33,34,35,36]. These guidelines suggest performing skin tests and if negative, drug provocation tests (DPT). However, there are still substantial differences in the management of hypersensitivity reactions to drugs around the world related to differential antibiotic prescribing patterns, differences in patient selection or demographic and genetic differences [37] as well as variations in β-lactam reagent availability. For example, currently in Europe, amoxicillin, and its combination with clavulanic acid, accounts for around 50% of hypersensitivity reactions to drugs, while in the United States, it is the second after phenoxymethylpenicillin [38] and cephalosporins as cefuroxime and ceftriaxone [39]. In Europe and Australia, selective aminopenicillin reactions and cross-reactivity between aminopenicillins and cephalosporins appear to be more prevalent than in the United States [8,40,41,42,43]. Additionally, the use of the skin test reagents for assessing β-lactam allergy such as benzylpenicillin and sodium benzylpenilloate are only available in Europe, but not in the United States. This heterogeneity in current practice highlights the necessity for standardizing protocols on the management of patients with suspected antibiotic allergy.

## 5. The Role of In Vitro Tests in Antibiotic Allergy De-labeling

In vitro tests are a potentially complementary procedure for the diagnosis of allergic reactions and the identification of culprit drugs (Figure 2). However, controversies exist on their diagnostic value in daily clinical routine, and only a few of them are licensed and/or show enough evidence for recommendation [44]. The application of the different in vitro tests depend on the mechanism involved in the reaction, and nowadays, the most commonly used and recommended ones are based on the determination of drug-specific IgE (sIgE), besides the activation of effector cells (basophils) for IgE-mediated reactions, as well as the proliferation of drug-specific effector cells (T-cells) for T-cell mediated reactions [45].

### 5.1. Drug-sIgE Determination

Immunoassays are the most common in vitro method for determining drug-sIgE, based on the use of a solid phase functionalized with drug-carrier conjugates to which drug-sIgE binds [46]. The most used commercial method is fluoroimmunoassay ImmunoCAP^®^ (Thermo-Fisher, Waltham, MA, USA), only available for some β-lactam antibiotic determinants (penicilloyl G, penicilloyl V, ampicilloyl, amoxicilloyl and cefaclor). This method is recommended for diagnosing β-lactam IgE-mediated reactions after negative skin test (ST) in order to avoid drug provocation test (DPT) or before ST in life-threatening reactions or high risk-patients [29,44,47,48,49]. However, ImmunoCAP^®^ sensitivity for β-lactams shows a low and variable sensitivity (0–50%) [47,48,49] that correlates with the severity of the clinical symptoms [47]. Studies lowering the threshold from 0.35 to 0.1 kUA/L show an increase in the sensitivity up to 85%, however, specificity decreases up to 54% specially for cases with tIgE > 200 kU/L [50]. Moreover, false positive results to penicillin V have been reported [51]. Finally, a recent study has shown the usefulness of ImmunoCAP^®^ diagnosing cefazolin allergy (currently for research use only), with a sensitivity and specificity of 49% and 94%, respectively [52].

Alternatives for drug-sIgE detection which are not available in commercial assays are in-house radioimmunoassay. Sensitivity for β-lactams ranges from 42.9% to 75% and specificity ranges from 67.7% to 83.3% [47,53,54,55]. However, this method is essentially applied for research purposes to study the immunological recognition of new chemical structures, different carriers and different solid phases [55,56,57,58,59,60,61].

Assays for detecting sIgE are recommended to be performed after no longer than 3 years following the reaction [44] due to drug-sIgE decreases with time [62,63,64].

### 5.2. Basophil Activation Test

Basophil activation test is a cellular test based on the determination of basophil activation in the presence of a stimulus/drug through the detection of activation markers by flow cytometry [65]. It is recommended as complementary in vitro diagnostic test in the evaluation of IgE-mediated allergic reactions [44], specially to avoid DPT in severe reactions or in the evaluation of allergy to antibiotics when no other in vitro test is available, such as for clavulanic acid, or fluoroquinolones [44,66,67].

Main limitations of the basophil activation test application in daily clinical routine are related to the lack of standardization between laboratories [65] and non-optimal sensitivity [44,65]. Sensitivity for penicillins ranges from 22% to 55% and for clavulanic acid up to 52.7%, both of them with a good specificity (79–96%) [28,68,69,70,71]. Moreover, recent studies have demonstrated that the basophil activation test is a promising technique, complementary to skin tests, for diagnosing allergy to amoxicillin–clavulanic acid [72,73], cefazolin (sensitivity rate up to 66.7%) [74] and 5-nitroimidazole (sensitivity rate up to 83.3%) [75].

On the other hand, sensitivity for fluoroquinolones ranges from 36% to 70%, depending on the drug tested, with a specificity of 90% [59,76,77,78], and a high negative predictive value that helps decide whether to perform DPT or not [79].

Technical considerations to be considered are (i) systemic steroids and immunosuppressors should be avoided before testing due to decrease of basophil response [80]; (ii) the test should not be performed during infection or active chronic inflammatory conditions [81]; (iii) the interval time between the reaction and the test should be no longer than 3 years [44] because of reported negativization of sIgE levels over time [62,63,64]; (iv) up to 10% of patients can be ‘non-responders’, and in these cases, basophil activation test results cannot be interpreted [65].

### 5.3. Lymphocyte Transformation Test

The lymphocyte transformation test is a cellular test to determine the proliferative response of drug-specific T-cells in the presence of the suspected drug(s) [44]. This test is recommended for evaluating T-cell mediated hypersensitivity reactions to drugs [44,82,83,84]; however, the unavailability of commercial tests and the lack of standardization between laboratories hamper its use for clinical routine [85]. Sensitivity and specificity are variable, depending on the drug and the clinical symptoms [44]. The highest sensitivity values are obtained for β-lactam antibiotics (58–88.8%), with a high specificity (85–100%) [82,83,84,86,87,88,89,90,91,92,93,94]. Sensitivity studies in children are limited, but a recent study has shown a sensitivity and specificity of 52.9% and 95.8%, respectively, for β-lactam allergy [95].

Regarding clinical manifestations, sensitivity and specificity are higher in MPE, FDE, AGEP and DRESS than in SJS/TEN [90,96]. Interestingly, lymphocyte transformation test results have shown a good correlation with the algorithm of drug causality for the epidermal necrolysis (ALDEN) score, suggesting its usefulness for diagnosing severe reactions [97,98]. Moreover, its use to evaluate drug cross-reactivity [99] and to confirm the culprit drug in SJS/TEN cases with multiple drug therapy should be highlighted [98].

Regarding technical considerations, there is no consensus on whether the lymphocyte transformation test should be performed in the acute or resolution phase of the reaction [91,100], with a recent study suggesting better results if the lymphocyte transformation test is performed during the recovery phase in DRESS cases [101]. On the other hand, different studies have achieved an improvement in lymphocyte transformation test sensitivity by methodological modifications based on the use of antigen-presenting cells [87], the inclusion of drug metabolites [102], the depletion of regulatory T cells [103] or the evaluation of different subpopulations of effector cells [90,103,104].

## 6. The Role of In Vivo Tests in Antibiotic Allergy De-Labeling

Currently, in vivo tests including skin tests and/or the drug provocation test (DPT) remain key for de-labeling antibiotic allergy (Figure 2).

Skin tests are generally well tolerated, easy to perform and not expensive [105]. Skin tests encompass the skin prick test and intradermal test. The skin prick test is carried out by pricking the skin percutaneously with a pricking needle on the volar aspect of the forearm. If this is negative, an intradermal test can be performed by injecting the drug intradermally, raising a small wheal of 3 mm in diameter. Reading should be conducted after 15–20 min for IRs and 24–72 h for non-IRs. A result will be considered positive when the size of the initial wheal increases by 3 mm more in diameter. For IRs, those that appear within the first hour after drug intake, a positive ST result translates in the vast majority into an IgE-mediated type reaction, in contrast to delayed-type reactions, which are frequently T-cell mediated [23].

General considerations are being aware of when performing STs and the best time to carry out them. A refractory period has been observed in the first 4–6 weeks after reactions, inducing a false-negative result due to depletion of mediators in recent anaphylaxis [106]. Moreover, they are not exempt from systemic reactions, and for some drugs, standardized non-irritant concentrations are not well defined [35].

The sensitivity, specificity and predictive values of skin tests depend on the antibiotic. While β-lactams skin tests have the best evidence for IRs with a specificity around 97% when all the haptens are collected together with benzylpenicilloyl, a minor determinant mixture, ampicillin and amoxicillin, for quinolones is specificity around 46.5% for all skin tests (skin tests and intradermal test) [49,107].

A systematic review and a meta-analysis were published in 2021 with the goal of analyzing the accuracy of diagnostic tests for penicillin allergy. A total of 105 studies, conducted on patients reporting a penicillin allergy and in whom skin tests were performed and compared to drug challenge results, were included. Skin tests showed a summary sensitivity of 30.7% (95% CI, 18.9–45.9%) and a specificity of 96.8% (95% CI, 94.2–98.3%) skin tests appear to have low sensitivity and high specificity. Projected predictive values mainly reflect the low frequency of true penicillin allergy [108].

When clinical history, skin tests and in vitro tests cannot confirm the diagnosis or are not available, a DPT is recommended. DPT is considered the gold standard for confirming or excluding the diagnosis of antibiotic allergy [29]. DPT consists of administering consecutive and increasing doses of a drug to verify tolerance. This approach is not recommended for patients with a history consisting of severe and life-threatening reactions such as anaphylaxis, shock, SJS, interstitial nephritis, hepatitis or hemolytic anemia, and requires a monitored clinical setting with rescue medications available in the event of a reaction [24]. DPT can also be considered for finding alternatives. However, DPT protocols are not standardized and vary widely among studies for IRs and NIRs in terms of dose steps and time intervals between incremental doses. Prolonged DPT regimens have higher negative predictive values than shorter ones although with more adverse effects and a greater impact on health and costs and the risk of developing antibiotic resistance by administering subtherapeutic doses over days [109,110].

Interestingly, Prieto et al. investigated the utility of DPT in a unique dose followed by regular treatment at home without previous skin tests in children reporting mild NIRs to β-lactams. Only 12.4% out of 194 children presented a positive DPT, and all of them were non-severe reactions. This assessment was able to de-label 87.6% of the patients safely and with less-consuming resources [95].

Devchand et al. recently validated an antibiotic allergy assessment tool that enables risk stratification based upon reported phenotype, which may aid in the identification of patients with low-risk phenotypes amenable to direct rechallenge [111].

Currently, allergy departments are overwhelmed with outpatient antibiotic allergy assessments, specifically penicillin tests. Looking for a patient-driven tool that could effectively reduce the high burden of patients falsely labeled as penicillin-allergic, Trubiano et al. developed and validated a penicillin allergy clinical decision rule. Patients reporting penicillin allergy were included and in vivo tests (skin tests and/or DPT) were performed in order to confirm or exclude the diagnosis. Multivariable analysis identified four features associated with a positive penicillin allergy test result. These features were summarized in the mnemonic questionnaire PEN-FAST. A cut-off of fewer than three points in PEN-FAST was chosen and validated for low-risk penicillin allergy patients, with a negative predictive value of 96.3%. PEN-FAST could identify the low-risk group at the point of care, and they would not need formal allergy testing [112].

## 7. Novel Approaches for De-labeling Antibiotic Allergy

Due to the impossibility for allergists to evaluate all patients labeled as allergic to penicillin mainly in circumstances such as the urgent need for antibiotic therapy, and as many hospitals lack access to allergy units, in the United States the allergy assessment has been attempted to extend to the emergency department, infectious diseases, internal medicine, pediatric and pharmacy specialists by using a multidisciplinary approach, risk stratification and Electronic Health Records (EHR) [113]. One such approach has been the training of pharmacists in skin testing to antibiotics, who de-labeled inpatients identified in the EHR as allergic to penicillins by using filters that included the current use of antibiotics such as carbapenems or monobactams, among other factors [114]. A revision of this protocol was the use of a clinical decision support tool incorporated into the EHR that linked all aztreonam orders for patients with a penicillin allergy to a pharmacist penicillin ST consultation, as aztreonam has widespread use in patients labeled as penicillin-allergic [115]. These inpatient programs allow both immediate access to allergological evaluations and consequently rapid change in antibiotic prescriptions. However, these settings have demonstrated many limitations [116] for penicillin allergy de-labeling, including time constraints, limitations in knowledge and comfort and competing patient.

The implementation of telehealth methods, including clinical informatics and artificial neural network [117], along with continued education of health care providers have potential to improve EHR documentation and communication, thereby advancing patient safety efforts [118].

## 8. The Challenge of Acceptance of De-Labeling among Patients

Among the difficulties we face nowadays in confronting the high burden of patients labeled as allergic to penicillin is the acceptance of de-labeling by the patient once the diagnosis of allergy has been ruled out by the allergological study. This frequently occurs in adult patients who were labeled as allergic in childhood. Throughout their lives, they have been receiving negative conditioning about avoiding antibiotics, and even though allergy has been ruled out and they are told that taking antibiotics is safe, they may be afraid of taking them. In fact, up to 41% of patients labeled as allergic to penicillins avoid them despite testing negative [119]. This fact highlights the importance of the need for early intervention, since an allergological study is frequently postponed because parents consider it uncomfortable or painful for their child [120]. As mentioned above, currently there is a debate about the duration of the DPT. Although prolonged DPT has been associated with negative adverse effects, it has been reported that patients who underwent DPT over several days were more convinced of de-labeling compared to those with a single dose [121].

## 9. Avoiding Re-Labeling Antibiotic Allergy

A major issue is the persistence of antibiotic allergy labels in EHR despite the diagnosis having been ruled out by an allergological study. This has been reported to occur in up to 51% of cases [122,123,124,125], and has been associated with incomplete clinical understanding of how to interpret negative allergological tests as well as failure to remove the allergy label altogether [119,123]. This particularly occurs in elderly patients with acute mental impairment or dementia living in long-term care facilities [126]. This persistence of the antibiotic allergy label despite an allergological study ruling it out leads to the use of alternative medication, with perioperative use of vancomycin and clindamycin reported in 18% of patients in whom β-lactam allergy having been ruled out by an allergological study [125]. This highlights the importance of both patient and clinicians being aware of the clinical significance of the results of the allergology study performed to rule out antibiotic allergy and in improving the integration of EHR across medical systems. A multidisciplinary approach including patient education, EHR alerts and wallet cards documenting penicillin allergy testing results has been found to reduce re-labeling [122]. However, more research is needed to prevent re-labeling.

## 10. Conclusions

Antibiotic allergies are an important barrier to effective antibiotic treatments as they have been associated with increased prescriptions of alternative broad-spectrum antibiotics, and consequently, with a high risk of antibiotic resistance.

However, many patients report an allergy to antibiotics, and frequently to penicillins, but only a small proportion have a true allergy. Therefore, an allergological evaluation is crucial for de-labeling antibiotic allergies and it is an effective tool for combating the use of broad-spectrum antibiotics as well as prevents the spread of antibiotic resistance. The standardization of skin testing and DPTs protocols is crucial in order to improve the diagnosis accuracy and patient safety. The use of reliable in vitro tests should be increased, especially for evaluating subjects who experienced severe reactions, in order to reduce the risk of systemic reactions to the in vivo tests. Moreover, there is an urgent need for developing new tools that can be used as screening and risk stratification for patients labeled as antibiotic allergic that will lead to easier and faster de-labeling in routine clinical practice. In addition, efforts have also been focused on designing methods for avoiding antibiotic allergy re-labeling.

## Figures and Tables

**Figure 1 antibiotics-11-01055-f001:**
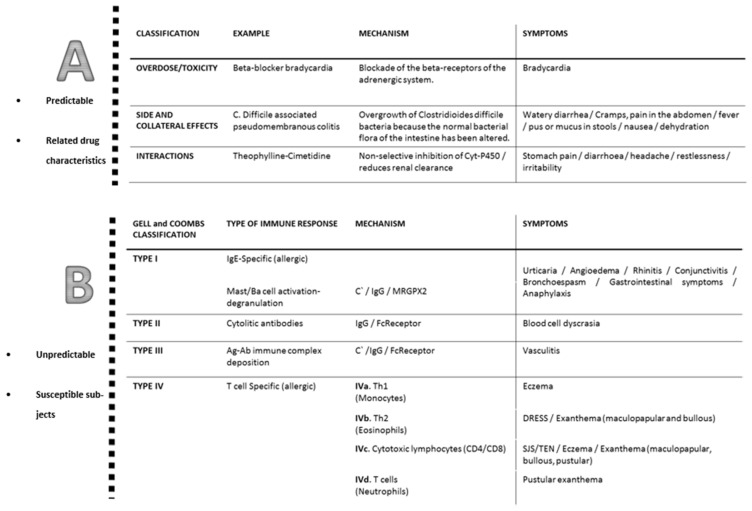
Classification of hypersensitivity reactions to antibiotics.

**Figure 2 antibiotics-11-01055-f002:**
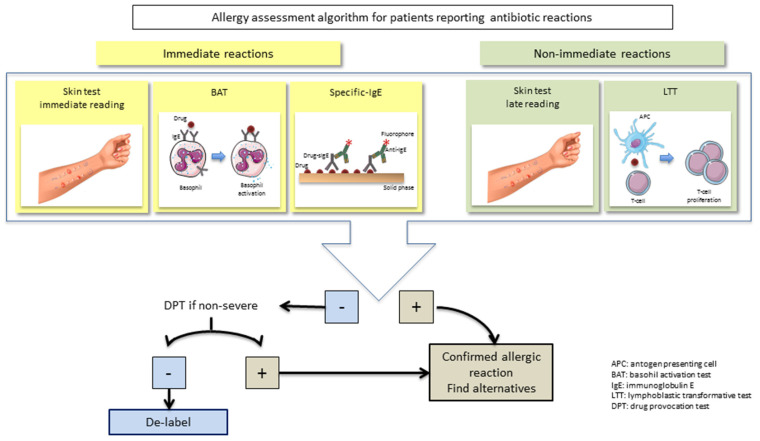
Algorithm for evaluating suspicion of allergic reactions to antibiotics.

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
