# Peer review of "Antibiotic Allergy De-Labeling: A Pathway against Antibiotic Resistance"

_antibiotics, 2022, doi:10.3390/antibiotics11081055_

Round 1
Reviewer 1 Report
Thanks for this manuscript. Antibiotic allergy de-labelling is imminent in optimizing antibiotic use. The text is well written, however I suggest some changes/extensions to improve the quality.
Content-wise:
- Paragraph 2: to the best of my knowledge, allergic reactions are more commonly classified in types I-IV then in A-B. There is certainly more than one way of classifying. Nevertheless, at present this paragraph is very confusing. A way to overcome this is to add the A-B classification to table 1 and highlight the overlap and differences.
- In clinical practice, antibiotic de-labelling is hampered by many practical aspects (underfunding, time constraints, lack of acceptation among patients and healthcare workers). Can you outline on this, and ways to overcome this? E.g. de-labelling in the emergency setting, rapid de-labelling clinic etc.
- Can you write a (short) section on the acceptance of de-labeling among patients?
Text/style:
- The introduction section is quite long and there is an element of repetition in lines 32-75. It would be good to write this more condensed.
- There are a lot of abbreviations in the text, some of which are not that common (e.g. BL; I would propose just writing beta lactams or HRD; hypersensitivity reaction). Please revise this and add a comprehensive list of all abbreviations that remain in the text.
- Please make sure that all name of bacteria are in italics.
Reviewer 2 Report
1. A separate ‘Introduction’ section should be added or section-1 can be renamed as Introduction.
2. The bacterial species name should be italic.
3. Few language errors are noticed. The manuscript should be proofread to correct those.
4. Figure-1 indicates the skin test. As the figure is referred in in vitro test, it indicates that skin test is an in vitro test.
5. It is an interesting review focusing on antibiotic allergies and the possibilities of de-labeling so that the use of broad-spectrum antibiotics could be minimized and drug resistance could be avoided. As the allergies are life-threatening, the possibility of de-labeling is remaining elusive.
6. Self-reported drug allergy is only applicable to a group of patients who are adults and able to communicate. In such patients, de-labeling can be useful. However, the patient groups who are not able to communicate such as pediatrics/geriatrics/mentally challenged /unconscious etc need an efficient test for allergy thus de-labeling is a challenge.
7. The major limitation of this manuscript is there are only possibilities and no definite solution. I would rather suggest translating these possibilities into a solution for the patient groups. Probably, segregating the patient groups and then the allergy tests that could be useful.
Round 2
Reviewer 1 Report
Thanks for this revision- it has really increased the quality of your manuscript. Yet, some minor parts are still quite wordy and may discourage the reader to go ahead (which would be a shame- as it is super interesting).
Therefore please consider:
* To delete lines 38-45; then the introduction will start spot-on with the topic, and it will directly be clear to the reader what the outline of the text is.
* Line 63: please merge citations.
* Line 64/65: ("higher costs ... healthcare system"): consider to write this as "higher healthcare expenditures".
* Line 76: please emphasize that only type B reactions are allergic reactions.
* Lines 93-96 ("type I ... mechanism"): please remove this- it is just a repetition of information that is in the table.
Author Response
- To delete lines 38-45; then the introduction will start spot-on with the topic, and it will directly be clear to the reader what the outline of the text is.
Thank you very much for this comment. We agree with the reviewer and we have deleted these lines.
- Line 63: please merge citations.
We thank very much this comment, we have put together the references.
- Line 64/65: ("higher costs ... healthcare system"): consider to write this as "higher healthcare expenditures".
We agree with the reviewer and we have made this change.
- Line 76: please emphasize that only type B reactions are allergic reactions.
Thank you very much for this comment, we agree with the reviewer and with this change we feel it is clearer for the reader.
- Lines 93-96 ("type I ... mechanism"): please remove this- it is just a repetition of information that is in the table.
Thank you very much for this comment. We agree with the reviewer and we have deleted these lines as the information is shown in Table 1.
